# Effects of breastfeeding on postpartum weight change in Japanese women: The Japan Environment and Children's Study (JECS)

**Masafumi Yamamoto[1], Mio Takami[1], Toshihiro Misumi[2], Chihiro Kawakami[3], Etsuko Miyagi🄾[4], Shuichi Ito[3], Shigeru Aoki🄾[1]\*, Japan Environment and Children's Study (JECS) Group¶**

**1** Perinatal Center for Maternity and Neonate, Yokohama City University Medical Center, Yokohama, Kanagawa, Japan, **2** Department of Biostatistics, Yokohama City University School of Medicine, Yokohama, Kanagawa, Japan, **3** Department of Pediatrics, Yokohama City University Graduate School of Medicine, Yokohama, Kanagawa, Japan, **4** Department of Obstetrics and Gynecology, Yokohama City University Graduate School of Medicine, Yokohama, Japan

¶ Membership of the Japan Environment and Children's Study is provided in the Acknowledgments.
* smyyaoki@yahoo.co.jp

**Data Availability Statement:** Data are unsuitable for public deposition due to ethical restrictions and legal framework of Japan. It is prohibited by the Act

## Abstract

### Aim

The aim of this study was to examine the relationship between breastfeeding and postpartum maternal weight change.

### Method

This study used data from the Japan Environment and Children's Study (JECS), an ongoing nationwide birth cohort study. Participants were categorized into two groups: full breastfeeding (FB) and non-full breastfeeding (NFB) groups. Postpartum weight changes between the FB (n = 26,340) and NFB (n = 38,129) groups were compared.

### Results

At 6 months postpartum, mean weight retention was significantly lower in the FB group than in the NFB group (0.2 vs 0.8 kg, respectively, p<0.001). Weight retention differed by pre-pregnancy body mass index (BMI), with postpartum weights of overweight (pre-pregnancy BMI 25.0–29.9) and obese (pre-pregnancy BMI ≥30.0) participants being lower than pre-pregnancy weight; this trend was more pronounced in the FB group than in the NFB group (overweight: −2.2 vs −0.7 kg, respectively; obese: −4.8 vs −3.4 kg, respectively). Factors affecting weight retention at 6 months postpartum were weight gain during pregnancy (β = 0.43; p<0.001), pre-pregnancy BMI (β = −0.147; p<0.001) and feeding method. FB resulted in lower weight retention than NFB (β = −0.107; p<0.001).

### Conclusion

Breastfeeding reduced maternal weight retention, which was greater in mothers who were obese before pregnancy. For obese women, active breastfeeding may improve their health.

on the Protection of Personal Information (Act No. 57 of 30 May 2003, amendment on 9 September 2015) to publicly deposit the data containing personal information. Ethical Guidelines for Medical and Health Research Involving Human Subjects enforced by the Japan Ministry of Education, Culture, Sports, Science and Technology and the Ministry of Health, Labour and Welfare also restricts the open sharing of the epidemiologic data. All inquiries about access to data should be sent to: jecs-en@nies.go.jp. The person responsible for handling enquiries sent to this e-mail address is Dr Shoji F. Nakayama, JECS Programme Office, National Institute for Environmental Studies.

**Funding:** This study was funded by the Ministry of the Environment, Japan. The findings and conclusions of this article are solely the responsibility of the authors and do not represent the official views of the above government. The funders had no role in study design, data collection and analysis, decision to publish, or preparation of the manuscript.

**Competing interests:** The authors have declared that no competing interests exist.

## Introduction

Breastfeeding has many established benefits for both mother and child. Its benefits for infants include reduced risk of infection, improved gastrointestinal function, and reduced risk of developing diabetes and obesity [1–4]. Breastfeeding has been reported to be beneficial for mothers in regard to stress tolerance and in reducing cardiovascular disease, diabetes, and hypertension [5, 6]. However, the role of breastfeeding in postpartum maternal weight management is not fully understood.

Some reports suggested that breastfeeding may promote postpartum weight loss [7–9], while others suggested it has no effect on maternal weight and body shape changes [10, 11]; there is no consensus. The contrasting findings may be due to differences in sample sizes and methods of assessing the duration and frequency of breastfeeding and weight change among the studies.

To address the sample size problem, this study used data from the Japan Environment and Children's Study (JECS), which surveyed approximately 100,000 pregnant women. The aim of this study was to examine the relationship between breastfeeding and postpartum maternal weight change.

## Materials and methods

This study used data from the JECS, an ongoing birth cohort study on the effects of antenatal and postnatal environmental factors on children's health and development. Enrollment in the JECS was conducted from January 2011 to March 2014. Approximately 100,000 children and their parents from 15 Regional Centers are enrolled in the JECS, and their health status is being followed from early pregnancy until children reach 13 years of age. The JECS also surveys lifestyle during pregnancy, the course of pregnancy and delivery, and the method of nutrition of the child after delivery. The present study examined the relationship between breastfeeding and maternal weight change after delivery from the following data: questionnaires from early pregnancy, third trimester of pregnancy, at the time of delivery, 1 and 6 months postpartum, and medical records at the time of delivery and 1 month postpartum.

The JECS protocol was reviewed and approved by the Ministry of the Environment's Institutional Review Board on Epidemiological Studies and the Ethics Committees of all participating institutions (Ethical Number: No.100910001). The present study was conducted with the approval of the Japan Environment and Children's Study Program Office. All procedures of this study were performed in accordance with the Declaration of Helsinki and with relevant guidelines and regulations. Informed consent was obtained in writing from all participants at the time of registration for JECS. If the participants were minor and unmarried, consent from a person with their parent or guardian was obtained. Details of the JECS protocol have been previously reported [12, 13].

### Study population

The present study was based on the dataset of jecs-ta-20190930, which was released in October 2019. The dataset included data on 103,060 pregnancies. The inclusion criteria were singleton pregnancies, full-term delivery, no underlying maternal disease, and no complications during pregnancy. The exclusion criteria were multiple pregnancies, preterm delivery (<37 weeks of gestation) and post-term delivery (≥42 weeks of gestation), miscarriages and stillbirths, underlying medical conditions and complications during pregnancy (hypertensive disorders of pregnancy, gestational diabetes mellitus, diabetes mellitus, thyroid disease, autoimmune disease, cardiac disease, renal disease, hepatitis, stroke, cerebral hemorrhage, epilepsy, blood disorders) and women with missing data. If a woman registered more than once, only the first

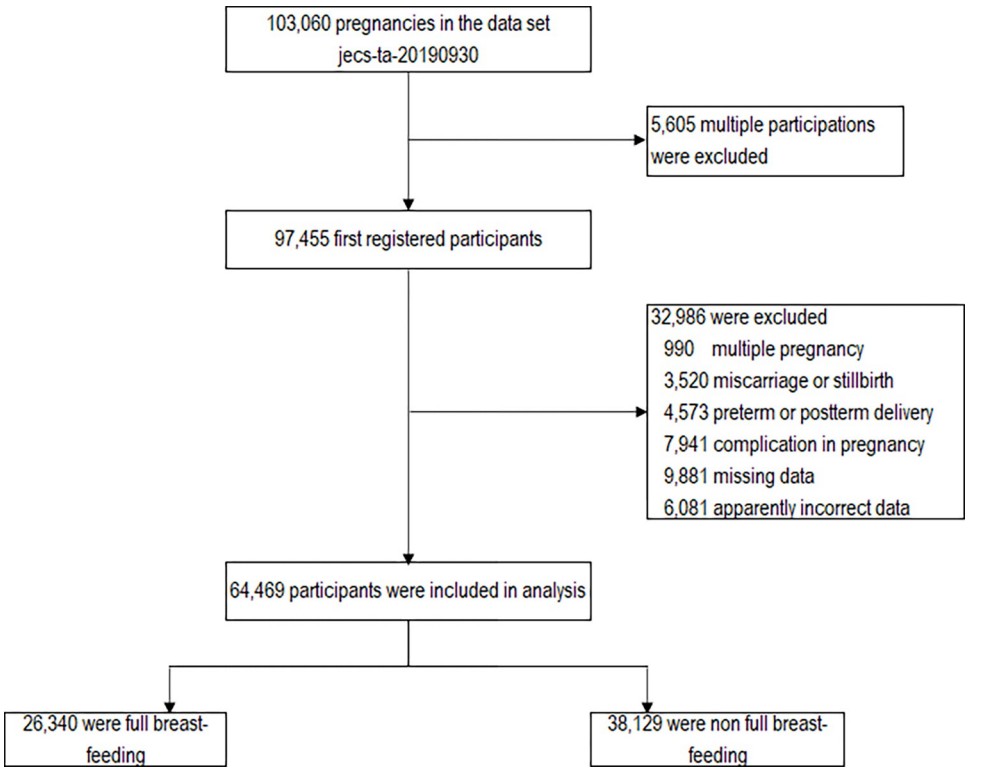

**Fig 1. Samples used in statistical analysis.**

registration was included in the analysis. The current study excluded those with endpoints outside the following ranges: height 120–200 cm, weight 25–200 kg, weight gain during pregnancy or post-partum weight loss −50 kg–+50 kg (defined as "apparently incorrect data", Fig 1). Using these criteria, 64,469 participants (age at delivery: 15–47 years old) were finally included in the analysis (Fig 1).

## Classification of population and outcomes

From the questionnaires of early pregnancy, second/third trimester of pregnancy, and at the time of delivery, the following data were collected: age, marital status, parity, smoking status, alcohol consumption, years of schooling, gestational age, mode of delivery, pre-pregnancy weight, and weight at delivery. From the questionnaire at 1 and 6 months postpartum, maternal weight was collected. The questionnaire at 6 months postpartum included a question on whether the baby has been breastfed or artificially fed every month for the first 6 months. From this question, those who fed only breast milk and never fed formula milk between 1 and 6 months postpartum were defined as the full breastfeeding (FB) group. Those who fed both breast milk and formula milk at any point between 1 and 6 months (i.e., mixed feeding), and those who fed only formula milk (i.e., artificial feeding), were defined as the non-full breastfeeding (NFB) group. To examine the relationship between the child's nutritional source and weight change by maternal body weight before pregnancy, subjects were divided into four groups on the basis of pre-pregnancy body mass index (BMI): underweight group ($<18.5$ kg/$m^2$), normal weight group ($18.5–24.9$ kg/$m^2$), overweight group ($25.0–29.9$ kg/$m^2$), and obese group ($\geq30.0$ kg/$m^2$).

The primary endpoints of weight loss after delivery and weight retention at 1 and 6 months postpartum were compared between the FB and NFB groups. This study also examined factors that influence postpartum weight changes using multivariate analysis.

## Statistical analysis

Weight loss after delivery was calculated by subtracting the body weight at 1 and 6 months postpartum from the body weight at delivery. Weight retention was calculated by subtracting the pre-pregnancy weight from the weight at 1 and 6 months postpartum. Pre-pregnancy BMI was calculated from pre-pregnancy weight and height. Weight gain during pregnancy was calculated by subtracting the pre-pregnancy weight from the weight at delivery. The appropriateness of weight gain during pregnancy was categorized into three categories following the Institute of Medicine (IOM) guidelines [14]: below, within, and above recommended weight gain management. The data were approximately normally distributed; therefore, Student's t-test and chi-square test were used to compare weight changes and participant characteristics between groups. Multiple regression analysis was used to assess important factors related to the change in postpartum weight. Method of breastfeeding, weight gain during pregnancy, pre-pregnancy BMI, age, primiparous, smoking during lactation, alcohol consumption during lactation, and years of education, all of which have been considered to affect weight change in previous studies [15–19], were explanatory variables. IBM® SPSS® Statistics for Windows, version 22 (IBM Corp. Armonk, N.Y., USA) was used for all statistical analyses. A value of p<0.05 was considered to be statistically significant.

## Results

Of the 64,469 participants analyzed, 26,340 (40.9%) and 38,129 (59.1%) were in the FB and NFB groups, respectively. Table 1 shows the background characteristics of the participants classified by pre-pregnancy BMI. In all BMI categories, participants in the FB group were less likely to be primiparous. Except for the obese group, the FB group participants were less likely to smoke, drink alcohol, and attend school for less than 12 years than those in the NFB group, and were more likely to be married. Except for the underweight group, pre-pregnancy BMI was smaller in the FB group than in the NFB group. Weight gain during pregnancy decreased as BMI increased (underweight 11.0 kg, normal weight 10.6 kg, overweight 8.4 kg, and obese 5.5 kg). In underweight and normal weight groups, weight gain during pregnancy was less in the FB group than in the NFB group; however, there was no significant difference in the other groups

Table 2 shows the maternal weight loss after delivery at 1 and 6 months postpartum. At 6 months postpartum, maternal weight loss was significantly greater in the FB group than in the NFB group for all BMI groups. At 1 month postpartum, except for the obese group, participants in the FB group lost approximately 0.1–0.3 kg more than those in the NFB group, but at 6 months postpartum, all BMI groups with FB lost significantly more weight than those with NFB, and there was a marked difference in weight loss (underweight 0.4 kg, normal weight 0.7 kg, overweight 1.4 kg, obese 1.0 kg).

Table 3 shows maternal weight retention at 1 and 6 months postpartum. The mean weight retentions of participants were 3.2 and 3.3 kg in the FB and NFB groups, respectively, at 1 month postpartum. The weight retentions at 6 months postpartum were 0.2 and 0.8 kg in the FB and NFB groups, respectively, almost back to the pre-pregnancy weight, which was significantly recovered with FB. Weight retention differed by pre-pregnancy BMI, and weight at 6 months postpartum was even lower than pre-pregnancy weight in the overweight and obese groups and was more pronounced in the FB group than in the NFB group (overweight: FB −2.2 kg, NFB −0.7 kg, obese: FB −4.8 kg, NFB −3.4 kg). The underweight group did not lose

**Table 1. Characteristics of participants classified by pre-pregnancy body mass index values.**

| | Maternal prepregnancy BMI category | | | | | | | | | |
|---|---|---|---|---|---|---|---|---|---|---|
| | Underweight | | Normal weight | | Overweight | | Obese | | All participants | |
| | FB<br>n = 4467 | NFB<br>n = 6180 | FB<br>n = 20190 | NFB<br>n = 28103 | FB<br>n = 1416 | NFB<br>n = 3090 | FB<br>n = 267 | NFB<br>n = 756 | FB<br>n = 26340 | NFB<br>n = 38129 |
| Age (year) | 30.3±4.72 | 30.2±5.08 | 31.1±4.70 | 31.4±5.09 | 31.5±4.75 | 31.8±5.09 | 31.5±4.73 | 31.8±4.99 | 31.0±4.71 | 31.3±5.11 |
| Height (cm) | 158.8±5.28 | 158.4±5.41 | 158.3±5.23 | 158.0±5.34 | 157.8±5.35 | 157.5±5.42 | 158.1±5.23 | 158.1±5.51 | 158.3±5.25 | 158.1±5.36 |
| Pre-pregnancy weight (kg) | 44.7±3.37 | 44.3±3.47 | 52.2±5.12 | 52.5±5.32 | 66.7±5.57 | 66.7±5.68 | 81.4±7.91 | 82.4±9.30 | 52.0±7.34 | 52.9±8.48 |
| BMI before pregnancy (kg/m2) | 17.7±0.67 | 17.6±0.72 | 20.8±1.58 | 21.0±1.65 | 26.7±1.33 | 26.9±1.38 | 32.6±2.46 | 32.9±2.84 | 20.7±2.66 | 21.2±3.13 |
| Weight at delivery (kg) | 55.5±4.73 | 55.4±5.05 | 62.8±6.20 | 63.2±6.50 | 75.0±6.85 | 75.2±7.20 | 86.7±8.59 | 88.1±9.75 | 62.5±7.70 | 63.4±8.66 |
| Weight gain during pregnancy (kg) | 10.9±3.26 | 11.1±3.52 | 10.6±3.49 | 10.7±3.74 | 8.4±4.45 | 8.5±4.85 | 5.3±5.13 | 5.6±5.19 | 10.5±3.61 | 10.5±3.96 |
| Gestational week at delivery (week) | 39.4±1.05 | 39.4±1.09 | 39.5±1.08 | 39.5±1.12 | 39.5±1.14 | 39.5±1.17 | 39.7±1.17 | 39.5±1.21 | 39.5±1.08 | 39.5±1.12 |
| Primiparous (%) | 1702 (38) | 3028 (49) | 7091 (35) | 12843 (46) | 386 (27) | 1153 (37) | 75 (28) | 268 (35) | 9254 (35) | 17292 (45) |
| Current marital status (%) | | | | | | | | | | |
| Married | 4258 (96) | 5702 (94) | 19394 (97) | 26571 (96) | 1373 (98) | 2944 (96) | 256 (97) | 718 (96) | 25281 (97) | 35935 (95) |
| Single (never married) | 140 (3.2) | 326 (5.3) | 558 (2.8) | 1034 (3.7) | 27 (1.9) | 95 (3.1) | 7 (2.7) | 25 (3.3) | 732 (2.8) | 1480 (3.9) |
| Divorced or Widowed | 24 (0.5) | 73 (1.2) | 81 (0.4) | 227 (0.8) | 5 (0.4) | 26 (0.8) | 1 (0.4) | 5 (0.7) | 111 (0.4) | 331 (0.9) |
| Smoking during breastfeeding period (%) | 114 (2.6) | 343 (5.6) | 356 (1.8) | 1238 (4.4) | 38 (2.7) | 189 (6.1) | 13 (4.9) | 54 (7.1) | 521 (2.0) | 1824 (4.8) |
| Alcohol intake during breastfeeding period (%) | 109 (2.4) | 285 (4.6) | 593 (2.9) | 1328 (4.7) | 40 (2.8) | 152 (4.9) | 7 (2.6) | 32 (4.2) | 749 (2.8) | 1797 (4.7) |
| Education (years) | | | | | | | | | | |
| <12 (%) | 170 (3.8) | 340 (5.6) | 516 (2.6) | 1248 (4.5) | 57 (4.1) | 188 (6.2) | 23 (8.8) | 58 (7.8) | 766 (2.9) | 1834 (4.9) |
| 12–14 (%) | 3093 (70) | 4471 (73) | 14178 (71) | 20615 (74) | 1095 (78) | 2410 (79) | 203 (78) | 604 (81) | 18569 (71) | 28100 (75) |
| >14 (%) | 1158 (26) | 1307 (21) | 5329 (27) | 5910 (21) | 255 (18) | 456 (15) | 36 (14) | 84 (11) | 6778 (26) | 7757 (21) |

Mean±SD.

BMI: body mass index, FB: full breast-feeding, NFB: non full breast-feeding, weight gain during pregnancy = (weight at delivery)-(pre-pregnancy weight).

more weight than their pre-pregnancy weight, and their weight retention was +1.0 kg, even if they were in the FB group.

Table 4 shows the results of multiple regression analysis. Comparison of method of breastfeeding, weight gain during pregnancy, pre-pregnancy BMI, age, primiparous, smoking during lactation, alcohol consumption during lactation, and years of education, showed that the greatest factor affecting weight retention at 6 months postpartum was weight gain during pregnancy (β = 0.43; p<0.001). The second most influential factor was pre-pregnancy BMI, with larger pre-pregnancy BMI leading to less weight retention (β = −0.147; p<0.001). The third most influential factor was the method of feeding the infant (FB versus NFB), with those in the FB group having less weight retention than those in the NFB group (β = −0.107; p<0.001). Similarly, weight loss after delivery at 6 months postpartum was influenced by weight gain during pregnancy, pre-pregnancy BMI, and method of breastfeeding, in that order. At 1 month postpartum, both weight retention and weight loss after delivery were influenced by weight gain during pregnancy, pre-pregnancy BMI, primiparous, and method of breastfeeding, in that order.

## Discussion

This study showed that FB resulted in significantly more postpartum weight loss than mixed feeding or artificial feeding. Weight loss was greater in the larger pre-pregnancy BMI, obese

**Table 2. Weight loss (kg) from delivery at 1 and 6 months postpartum.**

| Time point | | Feeding type | Maternal prepregnancy BMI category | | | | |
|---|---|---|---|---|---|---|---|
| | | | Underweight n = 10647 | Normal weight n = 48293 | Overweight n = 4506 | Obese n = 1023 | All participants n = 64469 |
| 1 month | Overall | FB | 6.9±2.06* [4467] | 7.2±2.33*** [20190] | 7.8±2.73** [1416] | 8.0±2.89 [267] | 7.2±2.32*** [26340] |
| | | NFB | 6.8±2.28 [6180] | 7.1±2.41 [28103] | 7.5±3.04 [3090] | 8.0±3.09 [756] | 7.1±2.47 [38129] |
| | Weight gain during pregnancy | | | | | | |
| | Below IOM recommendations | FB | 6.4±1.87*** [3272] | 6.6±2.18*** [12095] | 6.8±3.06** [487] | 7.3±2.74 [119] | 6.5±2.16*** [15973] |
| | | NFB | 6.2±2.03 [4329] | 6.4±2.16 [16391] | 6.3±3.52 [1072] | 7.1±2.66 [359] | 6.4±2.23 [22151] |
| | Within IOM recommendations | FB | 8.1±1.72 [1103] | 8.0±1.94*** [6827] | 7.8±2.20 [603] | 7.9±2.49 [95] | 8.0±1.94*** [8628] |
| | | NFB | 8.1±1.80 [1676] | 7.9±2.12 [9587] | 7.6±2.26 [1220] | 8.1±2.51 [213] | 7.9±2.10 [12696] |
| | Above IOM recommendations | FB | 10.2±3.21 [92] | 9.5±2.76 [1268] | 9.2±2.43 [326] | 9.9±3.06 [53] | 9.5±2.74 [1739] |
| | | NFB | 9.9±4.00 [175] | 9.5±2.84 [2125] | 8.9±2.73 [798] | 9.5±3.81 [184] | 9.4±2.96 [3282] |
| 6 month | Overall | FB | 9.9±3.42*** [4467] | 10.4±3.65*** [20190] | 10.6±4.57*** [1416] | 10.0±5.83* [267] | 10.3±3.71*** [26340] |
| | | NFB | 9.5±3.42 [6180] | 9.7±3.79 [28103] | 9.1±5.00 [3090] | 9.1±6.30 [756] | 9.6±3.92 [38129] |
| | Weight gain during pregnancy | | | | | | |
| | Below IOM recommendations | FB | 8.9±3.06*** [3272] | 9.0±3.21*** [12095] | 8.5±4.48*** [487] | 8.5±6.03* [119] | 9.0±3.26*** [15973] |
| | | NFB | 8.4±2.89 [4329] | 8.4±3.30 [16391] | 6.6±4.87 [1072] | 7.1±5.46 [359] | 8.3±3.39 [22151] |
| | Within IOM recommendations | FB | 12.4±2.59*** [1103] | 12.0±3.01*** [6827] | 10.8±3.94*** [603] | 9.5±4.37 [95] | 11.9±3.08*** [8628] |
| | | NFB | 11.8±2.88 [1676] | 11.2±3.21 [9587] | 9.4±4.07 [1220] | 9.1±5.29 [213] | 11.1±3.36 [12696] |
| | Above IOM recommendations | FB | 15.4±3.86 [92] | 14.6±4.12*** [1268] | 13.3±4.24*** [326] | 14.6±5.43 [53] | 14.4±4.21*** [1739] |
| | | NFB | 14.6±4.34 [175] | 13.8±4.36 [2125] | 12.0±4.77 [798] | 12.8±7.18 [184] | 13.3±4.73 [3282] |

Mean±SD, N in brackets

*p<0.05

**p<0.01

***p<0.001.

Weight gain during pregnancy = (weight at delivery)−(pre-pregnancy weight). BMI: body mass index, FB: full breast-feeding, NFB: non full breast-feeding, IOM: Institute of Medicine.

and overweight groups, whereas the underweight group did not lose more weight than their pre-pregnancy weight.

Full breastfeeding was significantly more effective in reducing weight retention and postpartum weight loss than mixed feeding or artificial feeding. Previous studies have found conflicting results on the effect of breastfeeding on maternal weight changes after delivery. These conflicting results may be due to differences in sample size, definition of breastfeeding, and timing of assessment of postpartum weight changes. Neville et al. reported in a review that almost half of the studies (10 of 21) that did not observe an association between breastfeeding and weight change tended to have a small sample size of less than 60 cases or short observation periods (≤3 months) [20]. There are several reports of an association between breastfeeding and postpartum weight change in studies with larger sample sizes and observation periods of more than 6 months postpartum. Baker et al. reported a prospective study of 3,630 cases that showed breastfeeding was effective in reducing weight retention at 6 months postpartum [21]. Krause et al. retrospectively reviewed 14,330 and 4,922 cases at 3 and 6 months postpartum, respectively [8]. This review found no significant association between breastfeeding method and weight retention at 3 months postpartum. However, at 6 months postpartum, the mean weight retention was reported to be 0.84 kg less for mixed feeding and 1.38 kg less for full

**Table 3. Weight retention (kg) at 1 and 6 months postpartum.**

| Time point | | Feeding type | Maternal prepregnancy BMI category | | | | |
|---|---|---|---|---|---|---|---|
| | | | Underweight n = 10647 | Normal weight n = 48293 | Overweight n = 4506 | Obese n = 1023 | All participants n = 64469 |
| 1 month | Overall | FB | 4.0±2.73*** [4467] | 3.4±3.04*** [20190] | 0.6±4.08** [1416] | -2.7±4.86 [267] | 3.2±3.21*** [26340] |
| | | NFB | 4.3±2.88 [6180] | 3.6±3.26 [28103] | 1.0±4.33 [3090] | -2.3±5.04 [756] | 3.3±3.53 [38129] |
| | Weight gain during pregnancy | | | | | | |
| | Below IOM recommendations | FB | 3.0±2.16*** [3272] | 1.9±2.36 [12095] | −3.0±3.18 [487] | −6.4±3.84 [119] | 1.9±2.65** [15973] |
| | | NFB | 3.2±2.26 [4329] | 1.9±2.46 [16391] | −2.7±3.27 [1072] | −5.7±3.78 [359] | 1.8±2.92 [22151] |
| | Within IOM recommendations | FB | 6.3±2.01** [1103] | 5.1±2.10*** [6827] | 1.2±2.32* [603] | −1.1±2.61 [95] | 4.9±2.47* [8628] |
| | | NFB | 6.5±1.98 [1676] | 5.3±2.25 [9587] | 1.5±2.47 [1220] | −0.9±2.60 [213] | 5.0±2.68 [12696] |
| | Above IOM recommendations | FB | 10.0±3.05 [92] | 8.5±2.73** [1268] | 4.9±3.11 [326] | 2.5±3.44 [53] | 7.7±3.34 [1739] |
| | | NFB | 10.7±3.16 [175] | 8.8±2.96 [2125] | 5.3±3.40 [798] | 2.7±4.31 [184] | 7.7±3.76 [3282] |
| 6 month | Overall | FB | 1.0±2.81*** [4467] | 0.2±3.21*** [20190] | −2.2±4.59*** [1416] | −4.8±5.67** [267] | 0.2±3.37*** [26340] |
| | | NFB | 1.6±2.82 [6180] | 0.9±3.50 [28103] | −0.7±4.83 [3090] | −3.4±6.40 [756] | 0.8±3.69 [38129] |
| | Weight gain during pregnancy | | | | | | |
| | Below IOM recommendations | FB | 0.5±2.68*** [3272] | −0.6±2.90*** [12095] | −4.6±4.23*** [487] | −7.5±5.77** [119] | −0.6±3.12*** [15973] |
| | | NFB | 1.0±2.45 [4329] | −0.1±3.06 [16391] | −3.1±4.22 [1072] | −5.8±5.93 [359] | −0.1±3.27 [22151] |
| | Within IOM recommendations | FB | 2.1±2.57*** [1103] | 1.1±2.95*** [6827] | −1.8±3.88*** [603] | −2.7±4.43 [95] | 0.9±3.14*** [8628] |
| | | NFB | 2.8±2.84 [1676] | 1.9±3.19 [9587] | −0.4±4.08 [1220] | −1.9±5.22 [213] | 1.7±3.41 [12696] |
| | Above IOM recommendations | FB | 4.9±3.60* [92] | 3.4±3.94*** [1268] | 0.8±4.38*** [326] | −2.2±4.64 [53] | 2.8±4.28*** [1739] |
| | | NFB | 5.9±3.71 [175] | 4.5±4.28 [2125] | 2.2±5.00 [798] | −0.6±6.89 [184] | 3.7±4.86 [3282] |

Mean±SD, N in brackets

*p<0.05

**p<0.01

***p<0.001.

Weight gain during pregnancy = (weight at delivery)−(pre-pregnancy weight). BMI: body mass index, FB: full breast-feeding, NFB: non full breast-feeding, IOM: Institute of Medicine.

breastfeeding than for full artificial feeding. In a recent retrospective study of 52,367 Taiwanese people, Waits et al. found that weight retention in the first 6 months after delivery was 0.7 kg less with full breastfeeding than mixed feeding, and 1.3 kg less with full breastfeeding than full artificial feeding [22]. The current study of 64,469 cases was the largest study described to date, and the subjects were divided into two groups: FB and NFB. As in a previous large study [22], FB resulted in significantly more postpartum weight loss and significantly less weight retention than NFB. Multiple regression analysis showed that weight retention and weight loss after delivery, at 6 months postpartum, were correlated with weight gain during pregnancy, pre-pregnancy BMI, and method of breastfeeding. These results showed that full breastfeeding strongly contributes to the promotion of postpartum weight loss and a reduction in weight retention.

Weight loss was greater in the larger pre-pregnancy BMI, obese and overweight groups, whereas the underweight group did not lose more weight than their pre-pregnancy weight. A previous meta-analysis examining the relationship between pre-pregnancy BMI, weight gain and weight retention reported that lower pre-pregnancy BMI and greater weight gain during

**Table 4. Factors affecting postpartum maternal weight.**

| Time point | Variable | Weight loss from delivery | | | | Weight retention | | | |
|---|---|---|---|---|---|---|---|---|---|
| | | B | β | 95% CI | p value | B | β | 95% CI | p value |
| 1 month | Weight gain during pregnancy | 0.331 | 0.524 | 0.326, 0.335 | <0.001 | 0.669 | 0.752 | 0.665, 0.674 | <0.001 |
| | Prepregnancy BMI | 0.185 | 0.228 | 0.179, 0.190 | <0.001 | -0.185 | -0.161 | -0.190, -0.179 | <0.001 |
| | Age | 0.017 | 0.034 | 0.013, 0.020 | <0.001 | -0.017 | -0.024 | -0.020, -0.013 | <0.001 |
| | Full breast-feeding | 0.248 | 0.051 | 0.215, 0.281 | <0.001 | -0.248 | -0.036 | -0.281, -0.215 | <0.001 |
| | Education | 0.035 | 0.007 | 0.000, 0.069 | 0.047 | -0.035 | -0.005 | -0.069, 0.000 | 0.047 |
| | Smoking during breastfeeding period | 0.135 | 0.011 | 0.048, 0.222 | 0.002 | -0.135 | -0.007 | -0.222, -0.048 | 0.002 |
| | Alcohol intake during breastfeeding period | 0.016 | 0.001 | -0.067, 0.099 | 0.70 | -0.016 | -0.001 | -0.099, 0.067 | 0.70 |
| | Primiparous | -0.590 | -0.121 | -0.624, -0.556 | <0.001 | 0.590 | 0.086 | 0.566, 0.624 | <0.001 |
| 6 month | Weight gain during pregnancy | 0.597 | 0.592 | 0.590, 0.604 | <0.001 | 0.403 | 0.430 | 0.396, 0.410 | <0.001 |
| | Prepregnancy BMI | 0.177 | 0.137 | 0.169, 0.186 | <0.001 | -0.177 | -0.147 | -0.186, -0.169 | <0.001 |
| | Age | -0.007 | -0.009 | -0.012, -0.001 | 0.013 | 0.007 | 0.009 | 0.001, 0.012 | 0.01 |
| | Full breast-feeding | 0.777 | 0.100 | 0.726, 0.828 | <0.001 | -0.777 | -0.107 | -0.828, -0.726 | <0.001 |
| | Education | 0.211 | 0.026 | 0.158, 0.264 | <0.001 | -0.211 | -0.028 | -0.264, -0.158 | <0.001 |
| | Smoking during breastfeeding period | 0.333 | 0.016 | 0.199, 0.467 | <0.001 | -0.333 | -0.018 | -0.467, -0.199 | <0.001 |
| | Alcohol intake during breastfeeding period | 0.035 | 0.002 | -0.092, 0.162 | 0.54 | -0.035 | -0.002 | -0.162, 0.092 | 0.59 |
| | Primiparous | -0.259 | -0.033 | -0.311, -0.207 | <0.001 | 0.259 | 0.036 | 0.207, 0.311 | <0.001 |

B: partial regression coefficient, β: standardized partial regression coefficient.

BMI: body mass index, CI: confidence interval for partial regression coefficient.

Weight gain during pregnancy = (weight at delivery)–(pre-pregnancy weight).

pregnancy was associated with greater weight retention [23]. The multiple regression analysis of the current study showed similar relationships for pre-pregnancy BMI, weight gain during pregnancy and weight retention. The weight gain during pregnancy was 8.4±4.73 kg for the overweight group and 5.5±5.17 kg for the obese group, both less than that in the other groups. This finding may reflect the fact that the overweight and obese groups were instructed not to gain too much weight during pregnancy compared with the other groups; higher pre-pregnancy BMI was associated with recommended lower weight gain during pregnancy. The combined effect of breastfeeding and controlling weight gain during pregnancy may have resulted in greater weight loss in the obese and overweight groups. While the obesity epidemic remains a problem, maternal thinness is also a problem [24]. It has been reported that 4.3% of women in the United Kingdom and 9.0% of women in China had a BMI below the World Health Organization definition of thinness (BMI of less than 18.5 kg/m$^2$) at their first medical visit after pregnancy [25, 26]. In a Japanese survey, the percentages of women aged 20–29 years and 30–39 years with a BMI of less than 18.5 kg/m$^2$ were reported to be 21.8% and 17.1%, respectively [27]. In the present study, 16.5% (10,647/64,469) of women had a pre-pregnancy BMI less than 18.5 kg/m$^2$. Such pre-pregnancy thinness may be a major health concern for mothers who may lose more weight with breastfeeding. However, in this study, the mean weight retention in the underweight group at 6 months after delivery was less with FB compared with NFB (1.0 kg vs 1.6 kg, respectively), but with both feeding methods there was no after delivery weight loss compared with pre-pregnancy weight. Even among participants in the underweight group who did not gain enough weight during pregnancy to meet IOM guidelines, the mean weight retentions at 6 months after delivery were 0.5 kg and 1.0 kg with FB and NFB, respectively. Considering these results and the benefits of breastfeeding to the infant, there is no reason to abandon breastfeeding for fear of maternal weight loss, even in thin pregnant women, at least in well-nourished developed countries.

It is reported that a high pre-pregnancy BMI can hinder the initiation and continuation of breastfeeding [28, 29]. In the present study, the percentages of full breastfeeding for the distinct pre-pregnancy BMI groups were 42.0%, 41.8%, 31.4% and 26.1% in the underweight, normal, overweight and obese groups, respectively, and the percentage of full breastfeeding was lower in the overweight and obese groups than in other groups. However, the current results show that the weight loss was greater in the obese and overweight groups. Therefore, there may be important clinical significance for breastfeeding in these groups. Professional education and support have been shown to be effective in promoting breastfeeding [30–32], and such support should be more actively provided to mothers with a high pre-pregnancy BMI. While breast-feeding has many advantages, there are women who have no choice but to use formula milk due to their complications [33]. In such cases, it is important to respect their decision and provide them psychological support and information on the alternatives to breastfeeding that can improve the health of the mother and child.

The strong feature of this study is the large sample size (64,469 cases). Although Japanese and other Asian women tend to be less obese in general, the large sample size allowed the current study to focus analysis on pregnant women with a high pre-pregnancy BMI. There are several studies on lactation in Japanese women. However, to the best of our knowledge, this is the first reported study on lactation and postpartum weight change in Japanese women. Conversely, there are two limitations to consider in this study. First, the JECS questionnaire does not include questions about diet and exercise after delivery, so this study could not assess the impact of changes in body weight because of energy intake or physical activity. The relationship between breastfeeding and postpartum weight change may be investigated in more detail by adding physical activity and energy intake to the study. Second, this study included participant responses to a questionnaire. There is a concern that self-reporting may lead to a decrease in measurement accuracy compared with direct measurement and data collection by medical staff.

## Conclusion

The present study showed that full breastfeeding produced significantly more postpartum weight loss than mixed feeding or artificial feeding. Weight loss was greater in the larger pre-pregnancy BMI, obese and overweight groups, whereas the underweight group did not lose more weight than their pre-pregnancy weight. Although a higher BMI before pregnancy tends to reduce initiation and continuation of breastfeeding, the results of this study should provide an incentive for pre-pregnancy obese mothers to initiate and continue breastfeeding.

## Acknowledgments

The authors would like to express our gratitude to everyone who participated in this study and to all those who were involved in the data collection. We would also like to thank the members of the JECS Group.

Members of the JECS Group as of 2021: Michihiro Kamijima (principal investigator, Nagoya City University, Nagoya, Japan), Shin Yamazaki (National Institute for Environmental Studies, Tsukuba, Japan), Yukihiro Ohya (National Center for Child Health and Development, Tokyo, Japan), Reiko Kishi (Hokkaido University, Sapporo, Japan), Nobuo Yaegashi (Tohoku University, Sendai, Japan), Koichi Hashimoto (Fukushima Medical University, Fukushima, Japan), Chisato Mori (Chiba University, Chiba, Japan), Shuichi Ito (Yokohama City University, Yokohama, Japan), Zentaro Yamagata (University of Yamanashi, Chuo, Japan), Hidekuni Inadera (University of Toyama, Toyama, Japan), Takeo Nakayama (Kyoto University, Kyoto, Japan), Hiroyasu Iso (Osaka University, Suita, Japan), Masayuki Shima (Hyogo College of

Medicine, Nishinomiya, Japan), Hiroshige Nakamura (Tottori University, Yonago, Japan), Narufumi Suganuma (Kochi University, Nankoku, Japan), Koichi Kusuhara (University of Occupational and Environmental Health, Kitakyushu, Japan), and Takahiko Katoh (Kumamoto University, Kumamoto, Japan).

We would also like to thank Charles Allan, PhD, from Edanz Group (https://jp.edanz.com/ac) for editing a draft of this manuscript.

## Author Contributions

**Conceptualization:** Masafumi Yamamoto, Mio Takami, Shigeru Aoki.

**Data curation:** Mio Takami, Chihiro Kawakami, Shuichi Ito, Shigeru Aoki.

**Formal analysis:** Masafumi Yamamoto, Toshihiro Misumi.

**Methodology:** Masafumi Yamamoto, Toshihiro Misumi.

**Supervision:** Chihiro Kawakami, Etsuko Miyagi, Shuichi Ito, Shigeru Aoki.

**Writing – original draft:** Masafumi Yamamoto.

**Writing – review & editing:** Masafumi Yamamoto, Mio Takami, Toshihiro Misumi, Etsuko Miyagi, Shigeru Aoki.

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
