## [Decision Letter · Decision Letter 0]

10 Feb 2022

PONE-D-21-38089Effects of breastfeeding on postpartum weight change in Japanese women: the Japan Environment and Children's Study (JECS)PLOS ONE

Dear Dr. Aoki,

Thank you for submitting your manuscript to PLOS ONE. After careful consideration, we feel that it has merit but does not fully meet PLOS ONE’s publication criteria as it currently stands. Therefore, we invite you to submit a revised version of the manuscript that addresses the points raised during the review process.

We look forward to receiving your revised manuscript.

Kind regards,

Linglin Xie

Academic Editor

PLOS ONE

Journal Requirements:

3. In your statement, please include the full name of the IRB or ethics committee who approved or waived your study, as well as whether or not you obtained informed written or verbal consent. If consent was waived for your study, please include this information in your statement as well.

 “This study was funded by the Ministry of the Environment, Japan. The findings and conclusions of this article are solely the responsibility of the authors and do not represent the official views of the above government.”

Reviewers' comments:

Reviewer's Responses to Questions

**Comments to the Author**

1. Is the manuscript technically sound, and do the data support the conclusions?

Reviewer #1: Yes

Reviewer #2: Yes

2. Has the statistical analysis been performed appropriately and rigorously? 

Reviewer #1: Yes

Reviewer #2: Yes

3. Have the authors made all data underlying the findings in their manuscript fully available?

Reviewer #1: Yes

Reviewer #2: Yes

4. Is the manuscript presented in an intelligible fashion and written in standard English?

Reviewer #1: Yes

Reviewer #2: Yes

5. Review Comments to the Author

Reviewer #1: The article demonstrates that full breastfeeding is beneficial in reducing postpartum weight compared to mixed feeding or artificial feeding. However, there are some parts that are unclear:

1. The article suggested in the discussion that the combined effect of breastfeeding and controlling weight gain during pregnancy could be more helpful for weight loss in overweight and obese population. However, Table 2 and 3 barely showed a significant effect of FB on weight loss and weight retention. Why will a combined method work?

2. In the discussion, the author indicates that the weight retention in the underweight group with FB is also lower than NFB; while the weight loss increased in the FB group. However, this means that with BF, underweight mothers lose more weight than NFB. Is this concerning? Will it be more beneficial for underweight mothers to gain weight postpartum?

3. Lastly, I’m wondering what if the author separates the NFB group into the mixed group and the artificial feeding group? Will the mixed feeding method show any beneficial outcome compared to both groups?

Reviewer #2: I really liked this article. I appreciate your team for investigating this important question. A couple of questions/suggestions.

1. In the methods section (line 97), the exclusion criteria for weight include those outside of the range 25-300kg. Is this the true range, or a typo? It seems that 300kg is extremely heavy.

2. Do you have any information about if the weight loss from the FB overweight and obese mothers lost enough weight to improve their pre-pregnancy BMIs? For example, was an FB obese mother's postpartum weight loss enough to improve their BMI to an overweight status?

3. Do you know if the women were able to retain their weight loss after they were finished breastfeeding? I'm wondering if the body condition of these mothers that lost a great deal of weight postpartum was retained even 1-2 years after delivery.

4. There is several instances of switching tenses (i.e. from third to first person point of view) in the writing style. Converting all of these sentences to read in third person would help the manuscript read better.

6. PLOS authors have the option to publish the peer review history of their article (what does this mean?). If published, this will include your full peer review and any attached files.

Reviewer #1: No

Reviewer #2: No

---

## [Author Response · Author response to Decision Letter 0]

28 Feb 2022

Thank you very much for your review of our manuscript (PONE-D-21-38089) that we sent on December 1, 2021. We wish to express our appreciation to the reviewers for their insightful comments on our paper. 

Responses to the Comments by the associate editor:

Reply: We have confirmed that our manuscript and file naming meet the specified format. However, we apologize if there are any errors.

Reply: Thank you for your comment. In the Materials and methods section, we mentioned: "Informed consent was obtained in writing from all participants at the time of registration for JECS"(lines 78-79). In addition, we added the following information in case the participants are minors: "If the participants were minor and unmarried, consent from a person with their parent or guardian was obtained."(lines 79-81).

3. In your statement, please include the full name of the IRB or ethics committee who approved or waived your study, as well as whether or not you obtained informed written or verbal consent. If consent was waived for your study, please include this information in your statement as well.

Reply: Thank you for your comment. In the Materials and methods section, we mentioned: " The JECS protocol was reviewed and approved by the Ministry of the Environment’s Institutional Review Board on Epidemiological Studies and the Ethics Committees of all participating institutions (Ethical Number: No.100910001)."(lines 73-75). As we replied in the previous comment, we had obtained written informed consent from all participants.

 “This study was funded by the Ministry of the Environment, Japan. The findings and conclusions of this article are solely the responsibility of the authors and do not represent the official views of the above government.”

Reply: Thank you for your kind suggestion. The funders had no role in our study, therefore we have added the statement; "The funders had no role in study design, data collection and analysis, decision to publish, or preparation of the manuscript." as you suggested (lines 84-85).

Responses to the Comments by the reviewer #1:

1. The article suggested in the discussion that the combined effect of breastfeeding and controlling weight gain during pregnancy could be more helpful for weight loss in overweight and obese population. However, Table 2 and 3 barely showed a significant effect of FB on weight loss and weight retention. Why will a combined method work?

Reply: We apologize for our insufficient explanation of combined effect. As shown in table 3, there was less weight retention in FB than in NFB in overweight and obese groups (overweight: FB −2.2 kg, NFB −0.7 kg, obese: FB −4.8 kg, NFB −3.4 kg). In addition, as shown in table2, FB had more weight loss than NFB (overweight: FB 10.6 kg, NFB 9.1 kg, obese: FB 10.0 kg, NFB 9.1 kg). It was reported in previous study that less weight gain during pregnancy leads to less weight retention. In present study, as shown in table1, overweight and obese had less weight gain during pregnancy than the other groups. Therefore, we discussed that breastfeeding and the restriction of weight gain during pregnancy resulted in more weight loss and less weight retention in overweight and obese women, which we described as a combined effect in the discussion (lines 238-249).

2. In the discussion, the author indicates that the weight retention in the underweight group with FB is also lower than NFB; while the weight loss increased in the FB group. However, this means that with BF, underweight mothers lose more weight than NFB. Is this concerning? Will it be more beneficial for underweight mothers to gain weight postpartum?

Reply: Thank you for your comment. In general, there is an increased risk of low birth weight and preterm delivery in prepregnant underweight women. We added this consideration because the risk of the following pregnancy will be more increased if prepregnant underweight women lose even more weight due to breastfeeding. However, on average, underweight FB women did not lose more weight than their pre-pregnancy weight, therefore we concluded that underweight women can breastfeed with no problems.

3. Lastly, I’m wondering what if the author separates the NFB group into the mixed group and the artificial feeding group? Will the mixed feeding method show any beneficial outcome compared to both groups?

Reply: Thank you for the suggestion. As suggested, we believe the results when NFB is divided into full formula milk and mixed feeding would be valuable. However, the participants who fed only formula was very few in JECS study, therefore we considered it inappropriate to analyze complete formulas and mixed feeding separately. We hope you understand this situation.

Responses to the Comments by the reviewer #2:

1. In the methods section (line 97), the exclusion criteria for weight include those outside of the range 25-300kg. Is this the true range, or a typo? It seems that 300kg is extremely heavy.

Reply: We appreciate helpful comments of Reviewer #2. The reviewer's comment is correct. We reviewed the subjects again and found that there were no participants over 200 kg in the analysis, therefore we revised the upper limit of the range to 200 kg (line 99).

2. Do you have any information about if the weight loss from the FB overweight and obese mothers lost enough weight to improve their pre-pregnancy BMIs? For example, was an FB obese mother's postpartum weight loss enough to improve their BMI to an overweight status?

Reply: We think it is natural for the reviewer to question this point. Although this study was able to show that breastfeeding resulted in greater maternal postpartum weight loss, the effect of weight loss was not sufficient to improve the BMI category. It would be more effective to use diet or exercise therapy in addition to breastfeeding.

3. Do you know if the women were able to retain their weight loss after they were finished breastfeeding? I'm wondering if the body condition of these mothers that lost a great deal of weight postpartum was retained even 1-2 years after delivery.

Reply: We agree that additional information on duration of breastfeeding effect as the reviewer suggested would be valuable. Regrettably, however, because the observation period of maternal weight is up to 6 months after delivery in JECS study, we are unable to do the additional analysis. We have included your point as a consideration for future study.

4. There is several instances of switching tenses (i.e. from third to first person point of view) in the writing style. Converting all of these sentences to read in third person would help the manuscript read better.

Reply: We apologize for the difficulty in reading the manuscript. Since our native language is not English, we have asked the experienced English-native editor to edit this manuscript again.

---

## [Decision Letter · Decision Letter 1]

21 Apr 2022

Effects of breastfeeding on postpartum weight change in Japanese women: the Japan Environment and Children's Study (JECS)

PONE-D-21-38089R1

Dear Dr. Aoki,

We’re pleased to inform you that your manuscript has been judged scientifically suitable for publication and will be formally accepted for publication once it meets all outstanding technical requirements.

Kind regards,

Linglin Xie

Academic Editor

PLOS ONE

Additional Editor Comments (optional):

Reviewers' comments:

Reviewer's Responses to Questions

**Comments to the Author**

1. If the authors have adequately addressed your comments raised in a previous round of review and you feel that this manuscript is now acceptable for publication, you may indicate that here to bypass the “Comments to the Author” section, enter your conflict of interest statement in the “Confidential to Editor” section, and submit your "Accept" recommendation.

Reviewer #1: All comments have been addressed

Reviewer #2: All comments have been addressed

2. Is the manuscript technically sound, and do the data support the conclusions?

Reviewer #1: Yes

Reviewer #2: Yes

3. Has the statistical analysis been performed appropriately and rigorously? 

Reviewer #1: Yes

Reviewer #2: Yes

4. Have the authors made all data underlying the findings in their manuscript fully available?

Reviewer #1: Yes

Reviewer #2: Yes

5. Is the manuscript presented in an intelligible fashion and written in standard English?

Reviewer #1: Yes

Reviewer #2: Yes

6. Review Comments to the Author

Reviewer #1: (No Response)

Reviewer #2: Thank you for submitting a revision to your manuscript and thank you for addressing the comments raised in the first submission. All of my comments have been addressed and I believe that it adds great strength to the manuscript. Hopefully in the future, there can be other studies that address this weight loss, specifically for the higher BMI groups, for long-term implications on their health.

7. PLOS authors have the option to publish the peer review history of their article (what does this mean?). If published, this will include your full peer review and any attached files.

Reviewer #1: No

Reviewer #2: No

---

## [Editor Report · Acceptance letter]

25 Apr 2022

PONE-D-21-38089R1 

Effects of breastfeeding on postpartum weight change in Japanese women: the Japan Environment and Children's Study (JECS) 

Dear Dr. Aoki:

I'm pleased to inform you that your manuscript has been deemed suitable for publication in PLOS ONE. Congratulations! Your manuscript is now with our production department. 

Kind regards, 

on behalf of

Dr. Linglin Xie 

Academic Editor

PLOS ONE